# Women's Experiences of Utilizing Religious and Spiritual Beliefs as Coping Resources After Miscarriage

**Felicity Agwu Kalu**

School of Nursing & Midwifery, Trinity College Dublin, 24 D'Olier Street, Dublin 2, Ireland; fkalu@tcd.ie

**Abstract:** Religious and spiritual beliefs enable some bereaved women to cope better with the devastating effects of the loss of a baby through miscarriage. The aim of this study is to explore the experiences of women's utilization of religious and spiritual beliefs as coping resources after miscarriage. Heideggerian interpretive phenomenological approach guided the study. Data were collected using audio-recorded, semi-structured interviews with 10 women and analyzed thematically using the method developed by van Manen. Data analysis revealed three themes: (a) feelings of grief (b) finding meaning in loss and the purpose of life and (c) religious and spiritual beliefs as coping resources. Religious and spiritual beliefs play a major role in the lives of many individuals, and act as coping resources for many women who have experienced a miscarriage. Bereavement care providers' understanding of religious and spiritual values of women who have experienced a miscarriage is crucial to meeting their spiritual care needs.

**Keywords:** miscarriage; pregnancy loss; bereavement; spirituality; spiritual care; coping

---

## 1. Introduction

The loss of a baby during pregnancy is a painful experience for many women (Murphy 2013; Bellhouse et al. 2018). Perinatal loss is an umbrella term used to describe miscarriage, stillbirth (fetal death) and death of a live born baby in the first twenty-eight days after birth (Corcoran et al. 2016; Sereshti et al. 2016; Agwu Kalu et al. 2018). Miscarriage is a spontaneous loss of a pregnancy before 24 weeks gestation while stillbirth is described as a baby born with no sign of life and known to have died from 24 completed weeks of pregnancy (RCOG 2010). The terminology and definition surrounding miscarriage and stillbirth vary with countries according to the number of weeks of the pregnancy (gestational age). For international comparability, the World Health Organization describes stillbirth as birth that occurred after 28 completed weeks of pregnancy (WHO 2016). Regardless of the gestational age, miscarriage is a global healthcare problem that can cause a psychological distress to many women (Kersting and Wagner 2012; Giannadrea et al. 2013; Olson 2013; Adeniran et al. 2015). It is estimated that one in four pregnancies end in spontaneous miscarriage (Olson 2013; Carolan and Wright 2017; Bellhouse et al. 2018). The effects of miscarriages and other pregnancy losses on women and their abilities to cope vary between different women, depending on the meaning their pregnancy, and pregnancy loss held for them (Shaw 2014; Golan and Leichtentritt 2016; Carolan and Wright 2017; Bellhouse et al. 2018). Although grief is a common experience, individuals respond in different ways to their losses because of religious beliefs, spiritual and cultural uniqueness of persons, and the ways by which grief is socially acknowledged (Shaw 2014; Allahdadian and Irajpour 2015). Grief reactions may include psychological, physical, social or emotional reaction to the loss (Mulvihill and Walsh 2013; Murphy 2013; Shaw 2014). Mothers and fathers grieve when they experience a perinatal loss but mothers tend to experience more intense grief than the fathers because of the physical and psychological

relationship of mother to the baby in utero (Murphy 2013; Mulvihill and Walsh 2013; Shaw 2014; Carolan and Wright 2017). In a qualitative study carried out in Ireland by (Mulvihill and Walsh 2013), it was found that women's husbands/partners appeared to get over the loss a lot quicker than the women. This was because although the men knew about the pregnancy, it took them a long time to know there was baby involved in the pregnancy whereas, for the women, pregnancy was all about the baby from the time they knew they were pregnant. The mothers felt they needed to protect their babies in utero and consequently blamed themselves following the loss of the pregnancy. In another qualitative study carried out by (Shaw 2014) in the UK with Pakistani Muslim women, it was reported that women felt distressed by their perceived failure to effectively protect the new life, as well as the failure of the healthcare professionals and society to recognise the personhood of their babies. In another qualitative study which was conducted in the USA, (Murphy 2013) found that some women's feelings after the loss of their babies was directly linked to societal ideals of womanhood, and perceived that they failed their babies by not understanding what was going on in their body.

Religious and spiritual beliefs play a major role in the lives of many people in Africa and worldwide (Shaw 2014; Arrey et al. 2016; Chitando 2017). In Nigeria, Christianity, Islam and traditional religions are the most commonly practiced religions (Yesufu 2016). In Nigeria and other Sub-Saharan countries, religion and spirituality help people to find meaning in life, and appreciate the value of life. Religion helps people to connect with their communities and God or the supreme being. Individuals are also enabled to cope and remain hopeful when faced with challenging situations (Arrey et al. 2016). In relation to maternity care, spiritual care and empathy enhances women's birth experience (Moloney and Susan 2015). Religious beliefs and practices irrespective of the type enable parents who have experienced a pregnancy loss not only to cope better with the devastating effects of the loss but also to adapt to their loss (Allahdadian and Irajpour 2015). In order to provide holistic, individualized and woman centred care to bereaved women and their families, maternity care providers and other bereavement care workers are expected to possess adequate knowledge and skills to meet the practical, emotional, socio-cultural needs, religious and spiritual needs of the women (Shaw 2014; Allahdadian and Irajpour 2015; Barry et al. 2017; Agwu Kalu et al. 2018). Having access to a dedicated perinatal bereavement support team can enhance the meeting of women's bereavement support needs (O'Connell et al. 2016). Although many women who experienced miscarriage or other pregnancy losses deeply grieve and mourn, such losses are often not always acknowledged by healthcare professionals and society because of insufficient understanding of the meaning of the loss for the woman and effects of the loss on her (Mulvihill and Walsh 2013; Golan and Leichtentritt 2016). In addition, some women find it uncomfortable disclosing their pregnancy with others earlier during the pregnancy because of the societal tradition of not discussing pregnancy until after 12 weeks (first trimester). This consequently leads to grieving in isolation and inadequate perinatal bereavement support to such women (Bellhouse et al. 2018). The aim of the study is to explore women's experiences of utilizing religious and spiritual beliefs as coping resources after miscarriages.

## 2. Materials and Methods

### 2.1. Design

A qualitative method using Heideggerian interpretive phenomenological approach guided the study. This approach facilitates a deeper understanding of the meaning of participants' lived experiences of the phenomenon (Heidegger 1962; Miles et al. 2013). Heidegger's philosophy focuses on ontology (the nature and relations of being). Heiddeger emphasizes the need for researchers to use their foreknowledge (preconceptions) to deduce meaning from their phenomena in order to improve or change practice (Crotty 1996). Using Heidegger's approach, the researcher was not concerned about bracketing of her preconceptions; rather she used her foreknowledge of providing perinatal bereavement care to parents and families to deduce meaning of women's experiences of utilizing

religious and spiritual beliefs as coping resources after miscarriages. Data were analysed thematically using the method developed by (Van Manen 1997) in order to locate meanings in the text.

### 2.2. Participants and Ethical Considerations

A purposive sample of 10 women who had experienced miscarriages 18–24 months prior to the study were recruited from a maternity unit in Eastern Nigeria. An administrative staff member in the maternity unit acted as a gatekeeper in the recruitment of the participants. The process of recruitment involved issuing prenotice/an expression of interest letter to participate in the study. This was followed by an information leaflet being issued subsequently. The gatekeeper was informed to circulate a participation information sheet to potential participants inviting them to participate in the study. The participants willingly participated in the study. They were chosen for the study because they had experienced miscarriages and were able to inform the study. These participants fitted the characteristics the researcher intended to study. The inclusion criteria were: women who were 18 years of age or older and women who had miscarriage 18–24 months prior to the study. Purposive sampling was employed in this study to gain rich information and an in-depth understanding of women's experiences of using religious and spiritual beliefs as coping resources after miscarriage. The time frame of 18–24 months was to give the women time to recover and at the same time being able to recall information about their miscarriages. Data saturation was achieved after the 10 interviews. That means that no new theme emerged at that point. Considerations to ethical issues were adhered to and permission to conduct the study was granted by Research Ethics Committee of the research site. Written informed consent was obtained from each participant by the researcher prior to the conducting of the study. Pseudonyms were used for the participants and all identifying information were removed from the transcripts prior to data analysis. The topic of miscarriage and bereavement in general may lead to a feeling of distress to some individuals. To safeguard the wellbeing of the participants, each of the participants was given contact details of bereavement support counsellor in case any participant required additional emotional support. The role of the bereavement support counsellor was to provide confidential one to one session to the participants if necessary. Although all the participants were given the contact details of the bereavement support counsellor, none of the participants reported the need for such support and did not avail of the counselling service offered.

### 2.3. Data Collection

Data were collected using face-to-face audio-recorded semi-structured interviews with 10 women who had experienced a miscarriage, at convenient locations identified by the participants. An interview guide with introductory open-ended questions such as, "Please can you tell me about the pregnancy you lost? What was it like being pregnant and losing the pregnancy? What did it mean losing the baby? How did you cope with the loss? What are the things that helped you cope with your loss? What are the things that did not help you cope with your loss? Have you any other thing to add? The rationale for using semi-structured interviewing was to give the women the freedom to respond in their own words and to provide as much details and explanations as they wished. The use of the semi-structured interviews helped the researcher to probe and clarify various points with the participants in a more flexible way. The interviews lasted between 50 and 90 min. The data were collected from January to February 2017 and later transcribed verbatim for the purpose of data analysis.

### 2.4. Data Analysis

The main purpose of data analysis in interpretive research is to locate meanings in the text (Van Manen 1997; Miles et al. 2013). The process of interpretation involved the researcher immersing herself in the data, engaging with it reflectively to identify themes, or meaning structures of the participants' lived experience of the phenomenon (Van Manen 1997). According to (Van Manen 1997), the analysis of a phenomenon involves trying to determine what the themes are, and the experiential structure that makes up the experience. After the transcription of the interviews, each transcript

was read several times to gain an understanding of each woman's story. Reading and re-reading the transcripts of the 10 participants, and also listening to the audio records, helped the researcher to become familiar with the data as the participants' thoughts, feelings and stories were going through the researcher's mind. The text was analysed by reflecting on it to find significant themes and at the same time engaging in the process of writing and rewriting. Initially each of the 10 individual interviews were analysed and written separately and the keywords and concepts were pulled out considering the parts as highlighted by van Manen. After that, the 10 interviews were analyzed together while looking for the themes and sub-themes and considering the whole as (Van Manen 1997) suggested. To make sense of the experiences of the women in this study, it was important for the researcher to move backwards and forwards in a hermeneutic circle as described by (Heidegger 1962) by reading and re-reading each of the transcripts, listening to the audio and analyzing the data over and over again. Each of the individual interviews as well as all the ten interviews together were analyzed. The transcripts were read and the words and sentences containing information relevant to the research questions were classified as key words, which were then condensed as concepts, and sorted into their respective sub-themes and themes. In the preliminary analysis, which involved analyzing each interview separately, or going from parts of the text to the whole (Van Manen 1997), the quotes were highlighted, copied and pasted, and became linked to the keywords, which led to the concept. In the final analysis, the highlighted statements became part of the sub-theme, and then to the theme. Two experts in phenomenological research were also asked to analyze and validate the interpretation of the themes.

## 2.5. Data Trustworthiness/Rigor

(Lincoln and Guba 1985) framework for ensuring rigor, which include credibility, transferability, dependability, and confirmability, guided the study. Credibility of the study was further achieved through reflexivity (Kalu and Bwalya 2017; Agwu Kalu et al. 2018). Credibility was established by using the participants for member checks in order to confirm the accuracy of the themes. The women were also asked to further explicate meaning of their experiences throughout the interview process. The use of two experts in phenomenological research to validate the interpretation of the themes further increases the credibility of the data. Transferability refers to how well the findings fit outside the study situation (Lincoln and Guba 1985). Qualitative studies are not aimed at generalization of the findings to wider population. The focus is more on the depth and richness of data and ensuring that the data have relevance when applied to other people, situations or context (Kalu and Bwalya 2017). To demonstrate the transferability of the findings, the researcher has placed the study in context by providing the readers with examples of other studies that described similar phenomena in the discussion section. Dependability is about the stability of research findings and the efforts of the researcher to appropriately account for any changing condition in the phenomena of study, design, or methodology (Lincoln and Guba 1985). The phenomenon of this study, the design, and methodology remained the same throughout the study. Confirmability refers to the steps the researcher took to demonstrate that the findings of the study emerged from the data and not her own dispositions (Lincoln and Guba 1985). Confirmability was achieved by ensuring the credibility, transferability and dependability of this study. Adequate account of the research process has been provided to ensure the confirmability of the study.

## 2.6. Reflexivity

The researcher's research knowledge and skills, her bereavement support knowledge and previous experiences of providing bereavement support in the clinical settings as a bereavement support midwife enabled her to become self-aware and be theoretically sensitized to the emergent data.

## 3. Findings

A total number of 10 women participated in the study. Their age profile ranged from 18–38 years. Other details of demographic characteristics of the study participants are presented in Table 1. Data analysis revealed three themes: (a) Feelings of grief (b) finding meaning in loss and the purpose of life and (c) religious and spiritual beliefs as coping resources.

**Table 1.** Demographic characteristics of the study participants.

| Characteristics of the Participants | N | % |
|---|---|---|
| Total number of women | 10 | 100% |
| **Marital status** | | |
| Married | 10 | 100% |
| Single | 0 | 0% |
| Separated/Divorced | 0 | 0% |
| Total | 10 | 100% |
| **Employment status** | | |
| Employed | 6 | 60% |
| Self-employed | 4 | 40% |
| Total | 10 | 100% |
| **Religion** | | |
| Christian | 9 | 90% |
| Muslim | 1 | 10% |
| Total | 10 | 100% |
| **History of miscarriages** | | |
| Number of women who had 1 miscarriage | 7 | 70% |
| Number of women who had 2 miscarriages | 2 | 20% |
| Number of women who had 3 miscarriages | 1 | 10% |
| Total | 100 | 100% |
| **Living children** | | |
| Number of women who had 1 or more than 1 living children at the time of interview | 10 | 100% |

### 3.1. Feelings of Grief

This theme describes the participants' grieving process and how they initially reacted to their loss. All the participants explained that they had lost something they really wanted. The following quotes from three participants serve as examples of the various feelings of women following their miscarriage:

'I felt numb initially. I couldn't believe it. I was angry and felt robbed of my baby and the mother-baby relationship I was looking forward to have with the baby. I was angry about the baby I lost'. (Linda)

'I was spotting from 7 weeks. I saw my doctor and he put me on bed rest. The bleeding stopped and I was very happy. I thought everything was okay. My happiness didn't last long because I lost the baby after 9 weeks. I was devastated because that was my third miscarriage. I thought everything was going well when the bleeding stopped. I blamed God for allowing this to happen to me the third time'. (Paula)

'I was bleeding heavily when I was 21 weeks pregnant. I was carrying two babies, twin pregnancy. It was hard to believe that I lost them. The doctor said the babies were too young to survive when they were born. They lived for a short time and died. I was heart broken and felt very helpless because I wasn't able to help them. I didn't want to let go of them but I knew I had to let go'. (Hannah)

### 3.2. Finding Meaning in Loss and the Purpose of Life

This theme describes how the participants' made sense of their loss as they grieved by searching for meaning of the loss and why it happened. All the participants reconstructed their personal world of meaning in relation to their miscarriage. In the absence of any known cause of their miscarriages, seven participants initially blamed themselves and God for the miscarriage. They eventually accepted the loss as the will of God. Two participants explained:

'I was very angry when it happened, when I lost the baby. I blamed my body for not keeping the baby. I really don't know why this happened. I had a scan a week before I miscarried and they said everything was good. I blamed God at the early stage for allowing the miscarriage to happen. After sometime, I didn't blame Him again. God knows the best. If it wasn't His will, this would not have happened. He knows why it happened. Who am I to question Him'. (Kate)

'My initial reaction was disbelief. I could not believe it. It took me a long time to get pregnant. When I saw the tiny baby, I couldn't believe how perfect he was. I thought about the miracle of God and life. He was a blessing to me. It was hard to accept the loss initially. I was heart broken. I eventually had to accept the loss as part of life. Everything is in God's hands'. (Liz)

All the participants spoke about the positive effects of midwives' bereavement support knowledge, skills and spiritual support and how these impacted positively on the abilities of the women to cope with their loss and adapt to the loss.

The midwife sat and listened to me. She was very knowledgeable, spiritual and caring. She asked me if I want to see the baby. I said yes and spent time with the baby. I have a good memory of her (the baby). That helped me to find ways to adjust and move on knowing that the baby will always be part of my family.'. (Mary)

### 3.3. Religious and Spiritual Beliefs as Coping Resources

This theme deals with how the religious belief of the women helped them to cope with their loss. The majority of the women described how their beliefs and connection with God or a higher spiritual being helped them to remain emotionally strong and hopeful of a better future in relation to having a successful pregnancy in the future. Spirituality and religious beliefs were the most common source of psychological support reported by the women in this study. The pastoral support received by the women also helped them to cope with their loss. This is evidenced in the excerpt below:

'I was loosing hope of getting pregnant again when I was mourning the loss of my baby. My pastor prayed with me. He told me that with God everything is possible. That gave me hope and emotional strength to move on'. (Anne)

Most women (*n* = 6) reported that prayers from family members also helped them to accept their loss, cope with the loss and adapt to the loss. This is illustrated in the following comments:

'After the third miscarriage at 9 weeks, I started loosing hope of having a baby but my sister kept reminding me to be hopeful in God. She has a strong faith and always prayed for me. She too had miscarriage before and later had two children. Her faith in God helped me to be emotionally and spiritually strong. I was pregnant again after 6 months and had a baby boy. He is full of life. It's good to have someone who can pull you up when your faith is down. My sister was always there for me. I thank God for her'. (Paula)

'My husband was very supportive. My faith was wavering after the miscarriage. I was sad and angry. I was afraid that I might loose another baby if I get pregnant again. We really wanted to have children. My husband didn't loose his faith. He kept praying. I joined him to pray when I could. I had no problem with my second pregnancy. Everything was fine. We have a baby girl and she is doing very very well. She is 9 months old now. I can't be thankful enough to God'. (Liz)

An aspect of religious and spiritual beliefs that enabled many of the women to cope with their loss was the comfort they derived from their belief in life after death and that their dead babies were in a safe place.

'As Christians we believe in life after death. Although I would have preferred not to loose the baby, I still believe he is in a good place. The angels are looking after him. That is what gives me consolation'. (Vicky)

Loosing a baby is very painful but I am sure my grandmother is looking after her, so she is in good hands, I'm okay she is in good hands. (Agnes)

Life is a gift from God. The Lord gives and also takes away. He is in the hand of God. He is okay. (Joy)

## 4. Discussion

Religious and spiritual beliefs play a major role in the lives of many individuals and act as a coping resource for many women who have experienced a miscarriage. The participants in this study tried to make sense of their loss and understand why it happened. Studies have found that making sense of a miscarriage or any type of loss is a way of approaching the overall process of meaning making (Golan and Leichtentritt 2016; Carolan and Wright 2017). In the current study, religious and spiritual beliefs had significant effects on how women grieved and accepted their loss. Women's spirituality enabled them to find meaning in their loss and purpose of life. For example, in the absence of known risk factors and in an attempt to find meaning in their loss, the majority of the women believed that it was the will of God that they experienced their miscarriage. This helped them to accept the loss. Similar to these findings, (Carolan and Wright 2017) reported that some bereaved families found comfort by defining the meaning of their miscarriage within the context of their spiritual and religious beliefs. (Allahdadian and Irajpour 2015) reported that religious beliefs and practices irrespective of the type enable bereaved parents find meaning in their loss, cope better with the devastating effects of the loss and also to adapt to their loss.

In this study, emotional and spiritual support from family members and health care professionals such as the midwives and pastoral care workers were vital in meeting the spiritual needs of the women. Several studies have shown that bereavement care providers need to be equipped with adequate perinatal bereavement support knowledge and skills for meeting the women's practical, emotional, socio-cultural, religious and spiritual needs of the women if holistic, individualized and woman centred care is to be provided to the women and their families (Shaw 2014; Allahdadian and Irajpour 2015; Barry et al. 2017; Agwu Kalu et al. 2018; Bellhouse et al. 2018).

## 5. Strengths & Limitations

The use of qualitative methods for this study allowed for an in-depth exploration of women's experiences of utilizing religious and spiritual beliefs as coping resources after miscarriage. The findings provide insights into the process of meaning making following miscarriage. The aim of this study was not to generalize the findings but to gain a deeper knowledge of the phenomenon being studied. Further studies are required to explore the experiences of the women's partners. In addition, quantitative study with larger numbers is required to see if the findings can be generalized to other populations.

## 6. Conclusions

Religious and spiritual beliefs play a major role in the lives of many Africans and act as a coping resource for many women who have experienced a miscarriage. This study has shown how women's spiritual and religious beliefs enabled them to find meaning in their loss and purpose of life as well as being able to cope with their loss. Bereavement care providers' understanding of the religious and spiritual beliefs and values of women who have experienced a miscarriage is vital to meeting the

spiritual care needs of these women. Women who have experienced a miscarriage should be facilitated to maximize the usage of their religious and spiritual beliefs as resources for coping and promotion of their overall sense of mental wellbeing.

**Funding:** This research was self-funded.

**Conflicts of Interest:** The author declares no conflict of interest.

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
