# Peer review of "Women’s Experiences of Utilizing Religious and Spiritual Beliefs as Coping Resources After Miscarriage"

_religions, doi:10.3390/rel10030185_

Round 1
Reviewer 1 Report
Dear authors,
the topic of your paper is of high interest and you provide a novel contribution, especially regarding Nigerian women.
I very much welcome concise presentations, yet they need to report enough information for scientific controllability, for the readers and for the scientific community.
In this regard, your paper is too short in a few points:
l. 57: A qualitative method using Heideggerian interpretive phenomenological approach guided the study. – Please give more information as this is not a standard approach
What were the questions of the semi-structured interview?
l. 81-103: was the data analysis conducted by only one researcher?
As to results, two categories are reported and illustrated by a few quotes. In sum, this seems to be a very slim result and in-depth-understanding would require more than the few remarks or comments on the quotes.
What is Heideggerian about the categories and results? Are there no more categories? How representative are the quotes for the ten participants?
These indications are meant to improve the quality of your paper as a scientific report on a very important topic.
It needs to meet the standards of the journal. Then it should be suited for publication in Religions.
Author Response
Dear Reviewer,
Thank you for the reviewing this manuscript. Your feedback have greatly helped to improve the quality of the paper.
I have provided more description of the research methods including Heideggarian phenomenological approach to enhance the readers’ understanding of the methods employed in this study and the appropriateness of these research methods.
The questions used during the semi-structured interviews have been included in the data collection section.
Data were initially analysed by one researcher. This was followed by further analysis and validation of the interpretation of the themes by two researchers who are experts in phenomenological research.
I have provided more information and details in the Findings/Results section. Three themes emerged from the data analysis. The quotes are representative of the participants’ comments discussed under the various sections. More comments from the participants have been added to enhance the readers’ understanding of how the themes emerged from the data analysis.
Finally, the manuscript has been proofread. Grammatical and typographical errors have been corrected.
Reviewer 2 Report
You have a good study, but much work is needed in as far as "fleshing out" your ideas, analysis and offering a comprehensive report of methods/outcomes. Please see the attached document with my comments/suggestions.

Author Response
Dear Reviewer,
Thank you for the reviewing this manuscript. You feedback have greatly helped to improve the quality of the paper.
I have revised the abstract and included sufficent background information, details and references in the introduction section.
I have provided more information in the materials and methods section (the research design, participants and ethical considerations, data collection and analysis, data trustworthiness/rigor reflexivity) to enhance the readers’ understanding of the methods employed in this study and the appropriateness of these study methods.
I have revised the findings/results section and included more details in this section. Three themes emerged from the data analysis. The quotes are representative of the participants’ comments discussed under the various sections. More comments from the participants have also been added to enhance the readers’ understanding of how the themes emerged from the data analysis.
The summary of the study has been presented in the conclusion section.
Finally the manuscript has been proofread. Grammatical and typographical errors have been corrected.
Round 2
Reviewer 1 Report
In my opinion, your additional work on the paper really improved it. I am looking forward to seeing it published. Congratulations!
Please allow two tiny remarks:
L 74f: correct the expression „often not always acknowledged” – you mean: frequently not acknowledged?
l. 130 – end mark of the quotes of the questions needed (i.e. “)
Reviewer 2 Report
Much improved from the original - there is clarity of thought and better writing style. Most notable, questions and gaps have been well attended to - the quality of work greatly improved.